# Omics Approaches in Uncovering Molecular Evolution and Physiology of Botanical Carnivory

**DOI:** 10.3390/plants12020408

**Published:** 2023-01-15

**Authors:** Anis Baharin, Tiew-Yik Ting, Hoe-Han Goh

**Affiliations:** Institute of Systems Biology, Universiti Kebangsaan Malaysia, UKM Bangi, Selangor 43600, Malaysia

**Keywords:** botanical carnivory, carnivorous plants, metabolomics, multi-omics, omics, proteomics, systems biology, transcriptomics

## Abstract

Systems biology has been increasingly applied with multiple omics for a holistic comprehension of complex biological systems beyond the reductionist approach that focuses on individual molecules. Different high-throughput omics approaches, including genomics, transcriptomics, metagenomics, proteomics, and metabolomics have been implemented to study the molecular mechanisms of botanical carnivory. This covers almost all orders of carnivorous plants, namely Caryophyllales, Ericales, Lamiales, and Oxalidales, except Poales. Studies using single-omics or integrated multi-omics elucidate the compositional changes in nucleic acids, proteins, and metabolites. The omics studies on carnivorous plants have led to insights into the carnivory origin and evolution, such as prey capture and digestion as well as the physiological adaptations of trap organ formation. Our understandings of botanical carnivory are further enhanced by the discoveries of digestive enzymes and transporter proteins that aid in efficient nutrient sequestration alongside dynamic molecular responses to prey. Metagenomics studies revealed the mutualistic relationships between microbes and carnivorous plants. Lastly, in silico analysis accelerated the functional characterization of new molecules from carnivorous plants. These studies have provided invaluable molecular data for systems understanding of carnivorous plants. More studies are needed to cover the diverse species with convergent evolution of botanical carnivory.

## 1. Introduction

Plants can adapt to harsh conditions by utilizing various means to obtain nutrients from natural habitats. Carnivorous or more specifically insectivorous plants can survive in a nutrient-deficient environment via morphological and physiological adaptations to a nitrogen-poor environment. Despite the fact that carnivorous plants have fascinated researchers and botanists for centuries, molecular studies of these plants have only been initiated in recent decades with the advent of molecular biology. There are over 700 species of carnivorous plants that can be found globally, which can be categorized into five orders [1]. Four of those orders have been studied through omics approaches with species from eleven genera and seven families (Table 1).

To date, there is no omics study on any carnivorous species of Poales order from the Bromeliaceae family. Carnivorous plants from Caryophyllales, Ericales, Lamiales, and Oxalidales often exhibit dramatic changes in leaf structure and morphology with specialized trapping mechanisms. They manifest carnivorous syndrome in scavenging available nutrients from various sources of nitrogen, including root uptake, predation, litterfall, atmospheric deposition, and defecation by animals for growth and survivability in unfavorable conditions [42]. Botanical carnivory includes prey attraction, capture, retention, digestion, and nutrient sequestration [43]. Some carnivorous plants produce secondary metabolites as pigmentation or aromatic compounds to attract prey and as a defense mechanism against microorganisms and oxidizing free radicals [44]. There are different types of specialized trap organs, either active or passive traps. For example, *N. rafflesiana* attracts prey by producing passive pitfall traps with bright-colored pitchers and aromatic nectars that mimic flowery scents to attract prey [45,46]. Once trapped, the plants will actively secrete hydrolytic enzymes to break down the prey and release nutrients for plant uptake [47]. Meanwhile, there are also reports of symbiotic relationships between carnivorous plants and microbes in nutrient assimilation [48].

Cost-benefit analysis of botanical carnivory has also been an important ecological topic of research interest to elucidate the evolution of carnivorous plants [49]. It has been reported that some carnivorous plants produce trapping organs of leaf origin with dual roles for carnivory and photosynthesis. At times when nitrogen deficiency becomes a limiting factor for survival, the function of the trapping organs will prioritize digestive enzyme secretion at the expense of reduced photosynthetic efficiency [43]. Most of the cost-benefit analyses are based on ecological studies. There is still limited omics research in this aspect with only a few exceptions, which will be discussed below, on the trap formation in *Nepenthes* species.

To our knowledge, this is the first comprehensive review of recent omics studies in carnivorous plant species which focuses on the evolution and mechanism of carnivory as well as molecular discovery and characterization such as digestive enzymes and metabolites. This review aims to serve as a reference and provide an overview of all omics studies towards a system understanding of botanical carnivory in diverse carnivorous plant species.

## 2. Omics Studies of Carnivorous Plants

Systems biology approach incorporates bioinformatics with different omics, such as genomics, transcriptomics, proteomics, metagenomics, and metabolomics [50]. Each omics approach focuses on profiling different molecules at different levels of the central dogma. Briefly, genomics identifies the pattern and composition of genome sequences coupled with bioinformatics analysis to predict gene functions. Transcriptomics profiling and quantification with RNA-sequencing (RNA-seq) can discover differentially expressed genes that could explain certain phenotypic traits with molecular physiology. Metagenomics analysis of nucleotide sequences from natural environments helps to elucidate the compositions and relationships of microbial communities with carnivorous plants. Proteomics identifies and quantifies all proteins in samples via mass spectrometry (MS) technologies, usually with sequence databases to study protein functions and features such as post-translational modifications and subcellular localization. Similarly, metabolomics detects, profiles, and quantifies changes in the metabolite constituent of samples based on compound databases or reference standards. Lastly, bioinformatics applies statistics and computational biology to analyze, integrate, and process biological data acquired from different omics. This review discusses the insights gained from different omics studies of carnivorous plant species.

## 3. Insights from Omics Studies

Extensive omics studies have been conducted to understand the molecular evolution of carnivory by studying the genomes and transcriptomes of Lentibulariaceae and Droseraceae family members. Furthermore, the candidate genes responsible for the development of the carnivory pitcher organ of *Nepenthes* species are revealed by transcriptomics studies. Another focus of the field is molecular physiology, in which the compositions of metabolites and proteins, especially the digestive enzymes and their dynamics in pitcher tissues and fluids, respectively, were investigated via metabolomics and proteomics informed by transcriptomics approaches. Furthermore, metagenomics studies were carried out in *Utricularia* and pitcher plants from the *Cephalotus*, *Nepenthes*, and *Sarracenia* genera to study the symbiotic relationships of microbial communities and the host carnivorous plants. Lastly, novel and potentially useful molecules were also characterized through in silico analysis complemented with omics data. The genome, plastome, and transcriptome studies are summarized in Appendix A for future reference. Findings from these studies will be discussed in the following sections.

### 3.1. Molecular Evolution of Carnivory Syndrome

Before the genomics era, studies of carnivorous plants were limited to phylogenetic analysis and genome size comparisons. The draft genomes of *Genlisea* [7] and *Utricularia* [51] from the Lentibulariaceae family allowed comparative genomic analysis of different carnivorous plant species for the first time. Based on flow cytometry analysis, these species have among the smallest angiosperm genomes, namely *G. margaretae* with 63 Mbp, G. *aurea* with 64 Mbp compared to the floating bladderwort (*U. gibba*) with 88 Mbp [6]. The distinction between these families was due to the variable gene mutation rates, with the *Utricularia* clade exhibiting higher mutation. In addition, comparative studies on the *U. gibba* and *U. reniformis* genomes showed differences in the repetitive sequences. The *U. reniformis* genome has a higher repetitive sequence (56%) than *U. gibba* does (32%) [9]. These species were hypothesized to downsize genome size via reducing the repetitive DNA, shrinking of non-coding DNA, or dispensing a proportion of non-coding sequences to undergo the diversification of angiosperms, which could lead to speciation with two different carnivorous syndromes between *Genlisea* (eel trap) and *Utricularia* (suction trap) from the same family in the aquatic habitats (Table 1). Through pairwise sequence alignment, it was revealed that both species have a different pattern of gene duplication. *U. reniformis* has higher tandem repeats and over-representation of gene ontology (GO) terms related to hydrolase activity, tropism, and a higher number of ABC transporters. These ABC transporters are essential to transport auxin and toxic compounds for terrestrial survival [52]. This phenomenon explains the way in which the species from the family Lentibulariaceae could colonize both the terrestrial and aquatic environments. Moreover, unique expression patterns of developmental genes in the terrestrial plant were also identified in *U. reniformis*, further supporting its classification as a terrestrial angiosperm [10]. The dynamic nature of genome with high gene family turnover rates is important for physiological adaptations [51,53]. In addition, studies also showed that *U. gibba* undergoes at least three rounds of whole-genome duplication (WGD), causing a drastic genome size difference from other species of the same clade [9]. Small-scale tandem duplications in *U. gibba* involved genes with the GO terms related to the transporters of oligopeptide, and dipeptide, cell wall (expansins, xyloglucan endotransglucosylases, class IV chitinases, and β-galactosidases), senescence-associated vacuolar cysteine proteases, and wax biosynthesis could give rise to carnivory in *U. gibba* [8].

Aside from *U. gibba*, a genome comparison study on the Venus flytrap (*D. muscipula*), *A. vesiculosa*, and *D. spatulata* showed WGD as the origin of carnivory-associated genes. This is followed by losses of genes involved in soil nutrition absorption as a major mechanism in the evolution of carnivory [11]. Different changes in genomic architectures in *D. muscipula*, *A. vesiculosa*, and *D. spatulata* eventually lead to different predatory mechanisms. For instance, transposon expansion in *D. muscipula* contributes to 38.78% of the genome size that might contribute to vesicle transport and mechanoelectrical signaling. *A. vesiculosa* diverged from *D. muscipula* after another event of WGD, whereas *D. spatulata* underwent tandem gene duplications event that might have derived the chemical sensing mechanism. These studies on carnivorous plant genomes provide a better understanding of the botanical carnivory evolution in the Droseraceae family [11]. In summary, these studies suggest that the changes in the genomic architecture of carnivorous plants such as genome size due to gene duplication and gene deletion events may result in the adaptive speciation of plant carnivorous morphology and physiology for survival.

Besides genome, the molecular evolution within the order of Caryophyllales has been explored through transcriptomics with targeted sequencing. For example, the transcriptomic data of *Dionaea muscipula*, *Drosera capensis* (Cape sundew), and *Nepenthes* species were analyzed to elucidate their evolutionary relationships, which showed that the clade contains at least seven independent paleopolyploidy events that lead to the speciation of these angiosperms within the same clade [2]. Furthermore, it has been recently reported that the post-speciation gene flow and introgression in c. 160 *Nepenthes* species are widespread throughout the rapid adaptive radiation that is only c. 5 million years old (Mya) based on the phylogenomic analysis of 235 double digest restriction site-associated DNA (ddRAD)-seq and 25 RNA-seq data [54].

On the other hand, the plastid genomes of *D. rotundifolia* and *Nepenthes × ventrata* from Caryophyllales have been studied to gain further insights into the molecular changes associated with the transition to carnivory. The plastome of *D. rotundifolia* is highly rearranged and has more repetitive sequences, while *N. × ventrata* plastome has a typical structure and gene context more related to angiosperms. The differences in plastome between *D. rotundifolia* and *N. × ventrata* could be due to the species representing different stages of evolution of plastome in carnivorous plants, in resemblance to that of parasitic plants transitioning from autotrophy to heterotrophic lifestyle. Such convergence may be driven by adaptation to obtain nutrients from other organisms, as in the case of carnivorous and parasitic plants [19].

In Droseraceae, comparative transcriptomics has identified the origin of plant carnivory syndrome. The origin of carnivory behavior in *D. muscipula* can be accounted for by at least three genes encoding FLYC1, FLYC2, and DmOSCA, which are homologs of mesenchymal stem cell-like (MSL) and OSCA/TMEM63 mechanosensitive ion channels [12]. FLYC1 transcripts are determined to be localized to mechanosensory cells of the touch-sensitive trigger hairs that are responsible to detect mechanical stimuli. FLYC1 protein can induce chloride-permeable stretch-activated current and trigger the following prey-capturing motion in *D. muscipula.* Furthermore, FLYC1 homologs (DcFLYC1.1 and DcFLYC1.2) were also determined to be expressed in *D. capensis* despite their difference in trap organs. Therefore, the authors proposed that the prey recognition mechanism in carnivorous Droseraceae arises from co-opting ancestral mechanosensitive ion channels to detect mechanical stimuli for prey capture [12].

Comparative proteomic analysis of the digestive fluids from *C. follicularis*, *D. adelae*, *N. alata*, and *S. purpurea* identified orthologous genes, such as GH19 chitinase, β-1,3-glucanase, PR-1-like protein, thaumatin-like protein, purple acid phosphatase, and RNase T2 [41]. Convergent amino acid substitutions were detected in GH19 chitinases, purple acid phosphatases, and RNase T2 homologs. The shared amino acid substitutions in RNase T2 between *Cephalotus* and the common ancestor of *D. adelae*, *D. muscipula*, and *N. alata* with different trapping mechanisms suggested that the convergent began before the diversification of trapping strategy [41]. Together, these studies have demonstrated different omics approaches in providing insights into the molecular evolution of botanical carnivory. Further details can refer to a recent review on this topic [1].

### 3.2. Evolutionary Development of Carnivory Organ

The development of pitchers has been studied through the transcriptomics approach. Analysis of the RNA-seq data indicated that the modification of the leaf into a pitcher is related to the altered expressions of leaf polarity genes, ASYMMETRIC LEAVES1 (AS1) and REVOLUTA (REV) as the genes were highly expressed in the tip of the leaf that later developed into a pitcher [30]. Furthermore, several candidate genes that might play a role in the development of the *Nepenthes* pitcher such as NkAS1 and NkREV were identified, leading to the proposal that NkAS1 inhibits lamina outgrowth and promotes the formation of the tendril, whereas higher NkREV expression is associated with pitcher formation in *N. khasiana* [30].

Comparative transcriptomics analysis of leaves and leaf-derived pitcher traps from *Nepenthes × ventrata* identified transcriptional signatures of the transition from leaf to pitcher organ, which is akin to the evolutionary origin of the floral organ [25]. In comparison with leaves, pitchers at all developmental stages were determined to be highly enriched with genes associated with shoot apical meristem (SAM) specification/organization, stress/defense response, flowering-related MADS-box transcription factors (TFs), biosynthesis of sucrose, wax/cutin, anthocyanins, and alkaloids that are similar with flower development, apart from digestive enzymes specific to carnivory syndrome. At the same time, the photosynthesis-related genes in pitchers were transcriptionally downregulated [25]. These transcriptomics studies showed the same developmental origins of carnivory and floral organs from leaves.

In Albany pitcher plant *C. follicularis,* flat leaf formation is favored at a low temperature of 15 °C while pitcher organs are predominantly formed at a higher temperature of 25 °C [41]. This phenomenon is studied through transcriptomics, in which RNA between the two different leaves was obtained and compared. The pitcher leaves showed differential expression of enriched GO terms of cell cycle and morphogenesis, whereas flat leaf transcriptome is enriched with GO terms related to photosynthesis.

### 3.3. Molecular Composition in Predatory Organs and Digestive Fluids

To date, the most extensive protein profiling studies of digestive fluids are reported in the *Nepenthes* species. The first proteome analysis of digestive fluids with relatively low throughput was reported 14 years ago in *Nepenthes alata*, which determined aspartic protease nepenthesins 1 and 2 to be the most abundant proteins apart from a few pathogenesis-related (PR) proteins, such as β-D-xylosidase, β-1,3-glucanase, and thaumatin-like proteins that inhibit microbial growth in pitcher fluids [55]. Later, the proteomics study of five *Nepenthes* species (*N. mirabilis*, *N. alata*, *N. sanguinea*, *N. bicalcarata*, and *N. albomarginata*) revealed 29 secreted proteins in pitcher fluids at different growth stages with 20 novel proteins, including serine carboxypeptidases, α- and β-galactosidases, lipid transfer proteins and esterases/lipases [24]. Aside from the PR proteins, three new nepenthesins (Nep3-5) and two novel prolyl proteases neprosin (Npr1-2) were discovered in *N. × ventrata* via proteomics informed by transcriptomics (PIT) approach with *N. rafflesiana* transcriptome sequences [23]. A novel aspartic protease Nepenthesin-6 (Nep6) was later reported based on quantitative PIT with Sequential Window Acquisition of All Theoretical Mass Spectra (SWATH-MS) using *N. × ventrata* transcriptome [31]. The pitcher fluid protein depletion experiment in *N. ampullaria* discovered new enzymes (xyloglucan endotransglucosylase/hydrolase proteins (XTH) and pectin acetylesterase (PAE) with possible functions in cell wall degradation that could contribute to the detritivorous habit of *N. ampullaria* [36].

Recently, a comparative PIT study was performed on *Nepenthes ampullaria*, *Nepenthes rafflesiana*, and their hybrid *Nepenthes × hookeriana* with different morphological traits and dietary habits to identify and compare proteins in pitcher fluids [37]. New proteins with diverse predicted functions such as amylase, invertase, catalase, kinases, ligases, synthases, esterases, transferases, transporters, and transcription factors. The digestive fluid protein compositions reflect the molecular physiology of pitchers from these *Nepenthes* species. The comparative analysis determined that transcripts and proteins identified in the hybrid *N. × hookeriana* resemble more of the insectivorous *N. rafflesiana* than the omnivorous *N. ampullaria*, which can derive nutrients from leaf litter. A metabolomics approach has been taken to study the effect of hybridization between the two aforementioned *Nepenthes* species with a consistent conclusion based on their chemical compositions and the putative identification of a flavonoid, astragalin as a distinctive molecular marker among the three species [22,35].

Among the different omics, metabolomic studies of carnivorous plants are relatively scarce. Previous studies mostly focused on the isolation of useful compounds [44] instead of metabolic profiling. The metabolic profiling of *Sarracenia* and *Darlingtonia* species identified unique compounds to differentiate between both genera and found coniine, which is a toxic compound previously isolated from *Sarracenia flava* [3]. A comparative metabolomics fingerprinting of lowland *N. ampullaria*, *N. rafflesiana*, and *N. northiana* compared to the highlander *N. minima* revealed species-specific temperature stress-induced metabolite markers and adaptive strategies as well as shared responses among the species [28]. Another metabolomics profiling study identified tissue-specific metabolite compositions the leaf blade and traps of *N*. *× ventrata* and changes upon prey digestion using MS and cheminformatics [33]. These studies highlight the challenges of metabolomics in putative metabolite identification due to high diversity of compounds and low abundance with many unknown metabolites yet to be identified from carnivorous plants.

On the other hand, 20 cDNAs of Sugars Will Eventually be Exported Transporters (SWEET) gene family were detected through transcriptomics analysis of *Nepenthes* leaf and pitcher at different stages of development [29]. The four classes of the SWEET family (class I–IV) are differentially expressed in mature leaf, primordial pitcher, immature unopened pitcher, and open pitcher, suggesting diverse gene functions. The expression data suggest that class I SWEET proteins are involved in the export of hexose sugar from the leaf, class I-III SWEET proteins involved in hexose and sucrose for pitcher growth in primordial and immature unopened pitcher stages, and lastly, class I, III, and IV SWEET proteins involved in the opened pitcher to transport sugars from the pitcher fluid to leaf. This study provides new insight into prey digestion in plant carnivory, in addition to other digestive enzymes such as chitinase, protease, and glucanase that are more well-studied. Overall, these newly identified proteins and metabolites shed light on the digestive mechanism of *Nepenthes* species.

In 2012, gel-based (SDS-PAGE and nLC-Q-TOF-MS/MS) and gel-free (HPLC-LTQ-Orbitrap XL MS) proteomics with transcriptomic approaches were used to identify secreted proteins in the digestive fluid of *D. muscipula* [56]. The study identified actively synthesized proteins in the trap, comprising chitinases, glucanases, nucleases, peroxidases, phosphatases, phospholipases, as well as proteases from the cysteine (4), aspartic (2), serine (1) families. Furthermore, VF chitinase-I was discovered through SDS-PAGE and MS analysis of the *D. muscipula* digestive fluid and expressed heterologously in *Pichia pastoris* showing the highest endochitinase activity at pH 5 and 50℃ [57]. On the other hand, the digestive fluid of *D. adelae* was analyzed using SWATH-MS for the identification of 26 proteins (19 novel proteins), with the majority being defense-related or pathogenesis-related (PR) proteins [58]. Six hydrolytic enzyme genes (Cysp1, da-I, Glu1, Hel1, Tlp1, and Chi1) were highly expressed at the head of glandular tentacles compared to other organs. These proteins are akin to that reported in *D. muscipula* and *Nepenthes* species. The chitinase-I was detected in both *D. adelae* and *D. muscipula* as secreted digestive protein. Like *D. muscipula*, the cysteine protease is the dominant digestive protease family in digestive fluid, in contrast to the aspartic family found in *Nepenthes*, *C. follicularis*, and *D. capensis* [56].

The high abundance of PR-related proteins in the digestive fluids of carnivorous plants from different genera including *Nepenthes*, *Dionaea* and *Drosera* suggests that the carnivorous capability evolved from the plant defense system [59]. Another evidence that plant carnivory evolved from plant defense is jasmonate (JA) signaling that is found in *Nepenthes*, *Dionaea* and *Drosera* genera. JA signaling plays important role in resistance towards biotic and abiotic factors, wound response, and defense against pathogen and insect infection in plant species. In these genera, the JA signaling is determined to be the signal transduction mechanism that converts prey-induced chemical (protein and chitin) or mechanical stimuli (action potential) to trigger transcription activation and synthesis of PR proteins, digestive enzymes, and secondary metabolites [59].

### 3.4. Molecular Dynamics in Carnivory Organ

Aside from discovering novel digestive enzymes, omics approaches were also used to study the carnivory behavior of carnivorous plants. RNA sequencing showed that *N. ampullaria* pitchers prioritize protein secretion rather than the expression of genes related to photosynthesis and metabolism after the loss of endogenous proteins in the pitcher fluids [34]. Furthermore, the study of the protein expression in pitcher organs in the early development has been conducted with the aid of RNA-seq and liquid chromatography-tandem mass spectrometry (LC-MS/MS) techniques to analyze the proteome of pitcher tissues and pitcher fluids of *N. × ventrata,* respectively [31]. The analyses demonstrate *N. × ventrata* pitcher organs continuously replenish protein in pitcher organs, especially proteins involved in plant carnivory evolved from a defense mechanism during early pitcher opening. Ultimately, this study provides insights into protein regulation during early pitcher opening before prey capture. Changes in the metabolite composition of pitcher tissues of *N. × ventrata* upon prey feeding have also been reported recently as mentioned above [33].

Furthermore, a 72 h metabolomic response of *D. capensis* simulated with prey capture was carried out to understand the plant physiology of the carnivorous plant by using ultra-high-performance liquid chromatography-tandem mass spectrometry (UHPLC-MS/MS) [15]. Furthermore, the identified metabolites were assigned to metabolic pathways and cross-compared with metabolites previously reported to be involved in carnivorous plants from different taxa. It is determined that secondary metabolites were highly expressed in comparison with unfed controls, especially 34 compounds that are also associated with carnivory in other species. Out of the 34 compounds, 11 compounds are exclusive to the Nepenthales order. More than 20 compounds showed 10-fold changes in concentrations, 12 of which with 30-fold changes are related to defense or attraction [15]. The metabolome response suggests that the role of secondary plant metabolites in plant carnivory could be overlooked in the past and more metabolomic studies are needed to investigate their roles in plant carnivory adaptation.

### 3.5. Symbiotic Relationships between Carnivorous Plants and Microbes

Metagenomics revealed a mutualistic relationship between carnivorous plants and the microbial community. For example, the analysis of metatranscriptomic data obtained from *U. australis* and *U. vulgaris* trap fluids showed that the trap environment can support microbial communities comprising bacteria, fungi, algae, and protozoa. At the same time, *Utricularia* provides a micro-ecosystem as a habitat with a complex food web that supports the establishment of the microbial community, which aids *Utricularia* in predation and sequestration of nutrients from the most diverse and abundant group, algae [60].

It is proposed that *Nepenthes* pitcher plants utilize a similar symbiotic relationship with the bacterial community in the pitcher fluid, as a metagenomics study of the pitcher fluid reveals 18 bacteria possess amylase, cellulase, chitinase, protease, and xylanase genes in their genomes [61]. Wet lab evidence proves that six of the bacterial chitinases exhibit chitinase activities. Therefore, this study supports the hypothesis that the bacterial community from *Nepenthes* pitcher fluid acts symbiotically with the host plant to enhance nutrient sequestration, although *Nepenthes* pitcher could self-sustain prey digestion with endogenous digestive enzymes [24].

Lastly, a metagenomics study on the bacterial and eukaryotic communities in *C. follicularis*, *Nepenthes*, and *Sarracenia* pitchers from Australia, Southeast Asia, and North America, respectively, showed similarities in the inquiline communities with specific bacterial taxa that are shared among all three independent lineages of these pitcher plants [40]. This suggests that the pitcher is the main selection factor for the microbiome, regardless of the geographical distribution or evolutionary history of the pitcher plants.

## 4. Molecule Characterization via Omics and In Silico Approaches

With advancements in computational technologies, multiple attempts using data from omics studies complemented with in silico methods could lead to the discovery and characterization of high-value molecules in carnivorous plants (Figure 1).

The conventional method of characterization of a molecule often involves PCR-based cloning techniques, recombinant expression in the host of choice, extraction and purification, activity or inhibitory assay, and crystallization, which are tedious and time-consuming. Since 2016, in silico approaches have been used for the characterization of novel chitinases [17], proteases from *D. capensis* [20,21], and more recently novel glutamic peptidase with post-proline cleaving activity, neprosin from *Nepenthes* species [38]. These studies often include analysis and annotation of omics data to identify target sequences, sequence alignment analysis, protein structure modeling, molecular dynamics simulation, and substrate docking. A pipeline consisting of omics and in silico analyses (Figure 1) could greatly reduce the effort and time required to identify and characterize molecules from carnivorous plants. This allows accelerated characterization of molecules from carnivorous plants that have commercial potential, especially proteases that have diverse functions in food processing and pharmaceutical industries. In the industrial settings, the proteases from carnivorous plants are sought after due to their substrate specificities, a wide range of working pH and temperature, and resistance against chemicals and denaturants [62].

## 5. Conclusions

Different omics approaches and methods provide systems biology insights into carnivory mechanisms in the diverse families of carnivorous plants. Single omics studies provide understanding at different molecular levels with different dynamics and regulations, which might not correlate with each other. For instance, genes subjected to transcriptional regulation, gene expression, and protein content might not be correlated due to post-transcriptional regulation, which in turn might not reflect the actual enzymatic activity because of post-translational regulation. On the other hand, metabolomic studies of carnivorous plants are lacking compared to the other omics. The combination of multiple omics is becoming more common with technological advancement and cost reduction in recent years. Multi-omics integration has discovered new molecular components in understanding the regulation of digestive enzyme secretion, especially in the Nepenthales order. More intensive data acquisition and analysis are still required for different orders of carnivorous plants. Nevertheless, the integration of different omics is the way forward for a holistic understanding of carnivorous plant systems biology.

## Figures and Tables

**Figure 1 plants-12-00408-f001:**
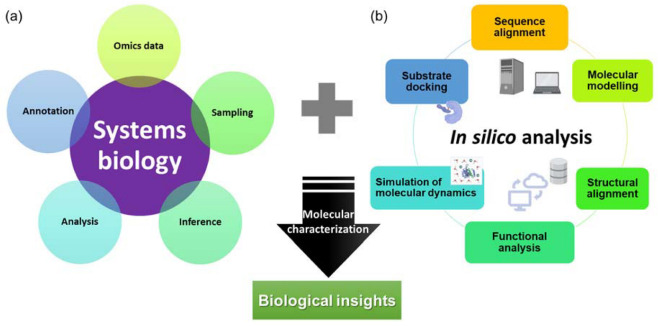
An overview of (**a**) systems biology and (**b**) in silico approaches for the characterization of novel and potential biotechnologically important molecules with biological insights.

**Table 1 plants-12-00408-t001:** The order, family, genus, and trap type of the carnivorous species discussed in this review.

Order	Family	Genus	Species	Trap Type	References
Caryophyllales	Drosophyllaceae	*Drosophyllum*	*D. lusitanicum*	Adhesive	[2]
Ericales	Sarraceniaceae	*Darlingtonia*	*D. californica*	Snap	[3]
		*Sarracenia*	*S. alata*, *S. flava*, *S. leucophylla*, *S. minor*, *S. oreophila*, *S. psittacina*, *S. purpurea*, *S. rubra*	Pitfall	[3,4,5]
	Lentibulariaceae	*Genlisea*	*G. aurea*, *G. hispidula*, *G. margaretae*	Eel	[6,7]
		*Utricularia*	*U. gibba*, *U. reniformis*	Suction	[6,8,9,10]
Nepenthales	Droseraceae	*Aldrovanda*	*A. vesiculosa*	Snap	[11]
		*Dionaea*	*D. muscipula*	Snap	[11,12,13,14]
		*Drosera*	*D. adelae*, *D. binata*, *D. capensis*, *D. rotundifolia*, *D. spatulata*	Adhesive	[2,11,12,15,16,17,18,19,20,21]
	Nepenthaceae	*Nepenthes*	*N. alata*, *N. albomarginata*, *N. ampullaria*, *N. bicalcarata*, *N. khasiana*, *N. minima*, *N. mirabilis*, *N northiana*, *N. rafflesiana*, *N. × hookeriana*, *N. × ventrata*	Pitfall	[2,19,22,23,24,25,26,27,28,29,30,31,32,33,34,35,36,37,38,39]
Oxalidales	Cephalotaceae	*Cephalotus*	*C. follicularis*	Pitfall	[40,41]

## Data Availability

Not applicable.

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
