# Peer review of "Omics Approaches in Uncovering Molecular Evolution and Physiology of Botanical Carnivory"

_plants, 2023, doi:10.3390/plants12020408_

Round 1

Reviewer 1 Report

This review article written by Baharin, Ting and Goh has been submitted to Plants, Special Issue: Advances in Carnivorous and Parasitic Plants and focuses on “Omics approaches in uncovering molecular evolution and physiology of botanical carnivory”. The review puts various omics approaches in the context of research that investigate the carnivorous syndrome in plants. It summarizes and highlights many studies that successfully used omics approaches to gain more insight in the molecular biology of carnivorous plants. Thus, this article provides a nice overview what has been achieved with omics up to now in carnivorous plants.

My main concern is that the metabolomics part is under-represented. There are two somehow hidden paragraphs related to metabolomics (lines 251-257 and 311-323) but these remained superficial and only three related articles are mentioned here although many more are listed in the References (3,15,16,22,28,33,35,44,45).

In some parts the review is written in such a way that I had the feeling that the authors wanted to convince the reader that omics approaches are important (e.g. chapter 2), which is not necessary. Also, figures 1 and 2 give the impression that they are presented for readers with little background knowledge. At least Figure 1 is not necessary. Instead, Figure 2 can be split into Figure 2a and 2b.

Minor points:

  1. Line 23-25, sentence hard to understand; please rephrase.
  2. Table 1, why Drosophyllaceae in the orders Caryophyllales AND Nepenthales?
  3. Line 42-43, sentence hard to understand; please rephrase.
  4. Table 2, please correct: N. ventricosa.
  5. Line 111 ff, given genome sizes, please cross check with table 2.
  6. Line 174-175, there must be more than only 3 genes necessary for the carnivory in Dionaea.

Author Response

Reviewer 1

This review article written by Baharin, Ting and Goh has been submitted to Plants, Special Issue: Advances in Carnivorous and Parasitic Plants and focuses on “Omics approaches in uncovering molecular evolution and physiology of botanical carnivory”. The review puts various omics approaches in the context of research that investigate the carnivorous syndrome in plants. It summarizes and highlights many studies that successfully used omics approaches to gain more insight in the molecular biology of carnivorous plants. Thus, this article provides a nice overview what has been achieved with omics up to now in carnivorous plants.

Thanks Reviewer 1 for appreciating this review and for constructive comments to improve this manuscript, which we have addressed in brown font color.

My main concern is that the metabolomics part is under-represented. There are two somehow hidden paragraphs related to metabolomics (lines 251-257 and 311-323) but these remained superficial and only three related articles are mentioned here although many more are listed in the References (3,15,16,22,28,33,35,44,45).

“Among the different omics, metabolomics studies of carnivorous plants are relatively scarce. Previous studies mostly focused on the isolation of useful compounds” (lines 254-255), which explains why the metabolomics part is under-represented. Based on the list of references in this review, we have added further descriptions for references 28 and 33 in the revised manuscript (Lines 254-266). Details as below:

  1.         Hotti, H.; Gopalacharyulu, P.; Seppänen-Laakso, T.; Rischer, H. Metabolite profiling of the carnivorous pitcher plants Darlingtonia and Sarracenia. PLOS ONE 2017, 12, e0171078, doi:10.1371/journal.pone.0171078. - Lines 253-256
  2.       Hatcher, C.R.; Sommer, U.; Heaney, L.M.; Millett, J. Metabolomic analysis reveals reliance on secondary plant metabolites to facilitate carnivory in the Cape sundew, Drosera capensis. Ann Bot 2021, 128, 301-314, doi:10.1093/aob/mcab065. - Lines 310-322
  3.       Mithöfer, A. A spotlight on prey-induced metabolite dynamics in sundew. A commentary on: 'Metabolomic analysis reveals reliance on secondary plant metabolites to facilitate carnivory in the Cape sundew, Drosera capensis'. Ann Bot 2021, 128, v-vi, doi:10.1093/aob/mcab093. - this commentary reference is removed
  4.       Rosli, M.A.F.; Azizan, K.A.; Baharum, S.N.; Goh, H.H. Mass spectrometry data of metabolomics analysis of Nepenthes pitchers. Data Brief 2017, 14, 295-297, doi:10.1016/j.dib.2017.07.068. - Data article of Ref. 35, Lines 250-253
  5.       Wong, C.; Ling, Y.S.; Wee, J.L.S.; Mujahid, A.; Müller, M. A comparative UHPLC-Q/TOF-MS-based eco-metabolomics approach reveals temperature adaptation of four Nepenthes species. Sci Rep 2020, 10, 21861, doi:10.1038/s41598-020-78873-3. - Lines 259-262
  6.       Dávila-Lara, A.; Rodríguez-López, C.E.; O'Connor, S.E.; Mithöfer, A. metabolomics analysis reveals tissue-specific metabolite compositions in leaf blade and traps of carnivorous Nepenthes plants. Int J Mol Sci 2020, 21, doi:10.3390/ijms21124376. - Lines 262-264, 319-321
  7.       Rosli, M.A.F.; Mediani, A.; Azizan, K.A.; Baharum, S.N.; Goh, H.H. UPLC-TOF-MS/MS-based metabolomics analysis reveals species-specific metabolite compositions in pitchers of Nepenthes ampullaria, Nepenthes rafflesiana, and their hybrid Nepenthes × hookeriana. Front Plant Sci 2021, 12, 655004, doi:10.3389/fpls.2021.655004. - Lines 250-253
  8.       Miclea, I. Secondary Metabolites with Biomedical Applications from Plants of the Sarraceniaceae Family. Int J Mol Sci 2022, 23, doi:10.3390/ijms23179877. - This is a review that described some of the secondary metabolites with biomedical applications rather than metabolomics analysis.
  9.       Di Giusto, B.; Bessière, J.-M.; Guéroult, M.; Lim, L.B.L.; Marshall, D.J.; Hossaert-McKey, M.; Gaume, L. Flower-scent mimicry masks a deadly trap in the carnivorous plant Nepenthes rafflesiana. Journal of Ecology 2010, 98, 845-856. - This is not a metabolomics study. 

In some parts the review is written in such a way that I had the feeling that the authors wanted to convince the reader that omics approaches are important (e.g. chapter 2), which is not necessary. 

Chapter 2 provides a background on different omics approaches for readers to comprehend the subsequent chapters/topics, which is explained by the last sentence of the paragraph as the main aim of this review. There is no intention of overemphasizing the importance of omics.

Also, figures 1 and 2 give the impression that they are presented for readers with little background knowledge. At least Figure 1 is not necessary. Instead, Figure 2 can be split into Figure 2a and 2b.

We have removed Figure 1 as suggested and revised Figure 2 to (a) and (b).

Minor points:

1. Line 23-25, sentence hard to understand; please rephrase.

The sentence has been revised to “ These studies have provided invaluable molecular data for systems understanding of carnivorous plants. More studies are needed to cover the diverse species with convergent evolution of botanical carnivory.”

2. Table 1, why Drosophyllaceae in the orders Caryophyllales AND Nepenthales?

We apologize for the mistake. Drosophyllaceae should be in the order of Caryophyllales only.

3. Line 42-43, sentence hard to understand; please rephrase.

We have revised the sentence to “To date, there is no omics study on any carnivorous species of Poales order from the Bromeliaceae family.” 

4. Table 2, please correct: N. ventricosa.

Table 2 is now a Supplementary Table 1 as suggested by Reviewer 1. The typo of ventricose has been corrected to ventricosa.

5. Line 111 ff, given genome sizes, please cross check with table 2.

The statement is based on the flow cytometry estimation of genome size, whereas Table 2 (now supplementary Table 1) is based on the latest genome analysis from sequencing.

6. Line 174-175, there must be more than only 3 genes necessary for the carnivory in Dionaea.

For clarity, we have revised the sentence to “ The origin of carnivory behavior in D. muscipula can be accounted for by at least three genes encoding FLYC1, FLYC2, and DmOSCA..”

Reviewer 2 Report

The manuscript of the review article “Omics Approaches in Uncovering Molecular Evolution and  Physiology of Botanical Carnivory ” by Anis Baharin et al. submitted to Plants presents the latest achievements of omics approaches in the physiology and evolution of carnivorous plants.

Due to the interesting subject and valuable overview of carnivorous plants physiology, the manuscript can be accepted for publication in Plants (special issue:     Advances in Carnivorous and Parasitic Plants) but after minor revision.

1. Figure 2 contains repetitions of Figure 1. Figure 1 is less informative, so it can be removed.

2.     Table 2 contains a lot of technical details not necessarily relevant to this review paper. I suggest moving it to supplementary materials.

3.     The part about the cost-benefits of carnivory is very sketchy, but this is an important aspect and is referred to later in the manuscript (e.g., trap-formation in Nepenthes).I suggest expanding on this part of the introduction. The aim of the study should be formulated in a separate paragraph.

4.     I suggest creating a Figure  that will contain the main topics of the work. This will tidy up the text of the manuscript and allow readers to follow the text more easily. The topics of chapters (phenomena that the work concerns): 3.1., 3.2., 3.3., 3.4., 3.5 can be placed in an exemplary model of a carnivorous plant.

Author Response

Reviewer 2

The manuscript of the review article “Omics Approaches in Uncovering Molecular Evolution and  Physiology of Botanical Carnivory ” by Anis Baharin et al. submitted to Plants presents the latest achievements of omics approaches in the physiology and evolution of carnivorous plants.

Due to the interesting subject and valuable overview of carnivorous plants physiology, the manuscript can be accepted for publication in Plants (special issue:  Advances in Carnivorous and Parasitic Plants) but after minor revision.

Thanks Reviewer 2 for appreciating this review with constructive comments for improvement, which we have revised accordingly in blue font.

  1. Figure 2 contains repetitions of Figure 1. Figure 1 is less informative, so it can be removed.

Fig. 1 has been removed as suggested.

  1. Table 2 contains a lot of technical details not necessarily relevant to this review paper. I suggest moving it to supplementary materials.

Table 2 is now the Supplementary Table 1.

  1. The part about the cost-benefits of carnivory is very sketchy, but this is an important aspect and is referred to later in the manuscript (e.g., trap-formation in Nepenthes).I suggest expanding on this part of the introduction. The aim of the study should be formulated in a separate paragraph.

In the revised manuscript, we have elaborated on the overview of cost-benefit analysis by adding the following statements: "Most of the cost-benefit analyses are based on ecological studies. There is still limited omics study in this aspect with only a few exceptions, which will be discussed below on the trap formation in Nepenthes species.” 

We also revised the aim of the study in a separate paragraph with an additional sentence: “This review aims to serve as a reference and provide an overview of all omics studies towards a systems understanding of botanical carnivory in diverse carnivorous plant species.”

  1. I suggest creating a Figure that will contain the main topics of the work. This will tidy up the text of the manuscript and allow readers to follow the text more easily. The topics of chapters (phenomena that the work concerns): 3.1., 3.2., 3.3., 3.4., 3.5 can be placed in an exemplary model of a carnivorous plant.

The different topics have been summarized in Figure 1 in the original manuscript for easy reference of readers but has been suggested to be removed by both reviewers. It can serve as a graphical abstract for an overview of this review. We think that the current manuscript structure with clear headings in the subtopics of chapter 3 should be sufficient to guide readers on the different topics covered in this review.